

# LTR retrotransposon-derived novel lncRNA2 enhances cold tolerance in Moso bamboo by modulating antioxidant activity and photosynthetic efficiency

Jiamin Zhao[1,*], Yiqian Ding[1,2,*], Muthusamy Ramakrishnan[3], Long-Hai Zou[1], Yujing Chen[1] and Mingbing Zhou[1]

[1] State Key Laboratory of Subtropical Silviculture, Bamboo Industry Institute, Zhejiang A&F University, Hangzhou, ZheJiang, China
[2] School of Forestry Science and Technology, Lishui Vocational and Technical College, Lishui, ZheJiang, China
[3] State Key Laboratory of Tree Genetics and Breeding, Co-Innovation Center for Sustainable Forestry in Southern China, Bamboo Research Institute, Key Laboratory of National Forestry and Grassland Administration on Subtropical Forest Biodiversity Conservation, School of Life Sciences, Nanjing Forestry University, Nanjing, Jiangsu, China
[*] These authors contributed equally to this work.

Corresponding author
Mingbing Zhou,
zhoumingbing@zafu.edu.cn

## ABSTRACT

In Moso bamboo, the mechanism of long terminal repeat (LTR) retrotransposon-derived long non-coding RNA (TElncRNA) in response to cold stress remains unclear. In this study, several *Pe-TElncRNAs* were identified from Moso bamboo transcriptome data. qRT-PCR analysis showed that the expression of a novel *Pe-TElncRNA2* in Moso bamboo seedlings reached its highest level at 8 hours of cold treatment at 4 °C and was significantly higher in the stems compared to the leaves, roots, and buds. Furthermore, cellular localization analysis revealed that *Pe-TElncRNA2* expression was significantly higher in the cytoplasm than in the nucleus. *Pe-TElncRNA2* overexpression in Moso bamboo protoplasts showed that *Pe-TElncRNA2* positively regulated the expression of *FZR2*, *NOT3*, *ABCG44* and *AGD6* genes. Further validation of this lncRNA in *Arabidopsis thaliana* enhanced antioxidant activities, as evidenced by increased superoxide dismutase (SOD) activity and proline content, as well as maximum photochemical efficiency PS II in dark-adapted leaves ($F_v/F_m$), in the transgenic plants compared to the wild-type controls. Conversely, malondialdehyde (MDA) content, a lipid peroxidation marker (a marker of oxidative stress), was significantly reduced in the transgenic plants. Notably, the expression levels of both *Pe-TElncRNA2* and the genes that were regulated by this lncRNA were upregulated in the transgenic plants after two days of cold stress treatment. These findings elucidate the critical role of LTR retrotransposon-derived lncRNAs in mediating cold stress responses in Moso bamboo.

# INTRODUCTION

Moso bamboo (*Phyllostachys edulis*), renowned for its rapid growth, is a versatile resource with applications in construction, textiles, biofuels, and food. Its potential to address food,

energy, and climate challenges has garnered significant attention (*Chen et al., 2022*; *Dlamini et al., 2022*; *Ramakrishnan et al., 2020*). While Moso bamboo dominates China's bamboo cultivation landscape, accounting for approximately 73.76% of its bamboo-growing areas, its productivity is hindered by abiotic stresses (*Wang et al., 2022*; *Zhu et al., 2024*). Among different stresses, cold stress induces oxidative stress and reduced cellular metabolism, ultimately inhibiting plant growth, including Moso bamboo (*Kim et al., 2024*; *Wang et al., 2022*). For example, during the rapid growth period, cold conditions affect internode length by slowing the growth of the shoots compared to warm conditions. Cell division-related genes (such as *PeCYC1BAT*, *PeCYCB2;4*, *PeCDC20.1*, *PeMAD2*, *PeTPX2*, and *PeTCX2*) and cell elongation-related genes (such as *PeEXPA1*) were upregulated with increasing temperature, suggesting that cold temperatures inhibit cell division-related genes during the rapid growth period in Moso bamboo (*Chen et al., 2022*).

In addition to identifying both cold resistance and cold susceptible genes (*Chen et al., 2022*; *Wang et al., 2022*), several studies have also identified genes in Moso bamboo, including *PeDREB1A*, *PeDREB2A*, *PeLEAs*, P*eAAAP*, *PeAQPs*, *PeTIFY*, *PeIQD*, *PeDi19-4*, and *PeZEP*, that exhibit differential expression in response to various abiotic stresses (*Huang et al., 2016a*; *Huang et al., 2016b*; *Liu et al., 2017*; *Liu et al., 2020*; *Lou et al., 2017*; *Sun et al., 2016*; *Wu et al., 2015*; *Wu et al., 2018*; *Wu et al., 2016*; *Wu et al., 2017*). Additionally, non-coding RNAs have been shown to regulate gene expression in response to abiotic stresses (*Ding et al., 2022*; *Yu, Ding & Zhou, 2023*). These findings collectively underscore the complexity of Moso bamboo's adaptive mechanisms to abiotic stresses. Although significant progress has been made in understanding the molecular responses of Moso bamboo to various abiotic stresses, the role of long non-coding RNAs (lncRNAs) in response to cold stress remains elusive. Therefore, a deeper understanding of cold stress response is still essential.

LncRNAs, which are more than 200 nucleotides long and have no translation potential, regulate many biological processes and function as either *cis* or trans regulators (*Zhou, Zheng & Wu, 2024*). They also activate chromatin remodeling, modulate alternative splicing, and influence post-transcriptional regulation. LncRNAs are associated with various agricultural traits and offer opportunities to be utilized as biomarkers and in breeding applications, being influential without protein production (*Gonzales et al., 2024*). LncRNAs exhibit differential expression in response to stress and regulate cold stress through various mechanisms (*Zhao et al., 2024*).

For instance, *COLD INDUCED lncRNA 1* (*CIL1*) acts as a positive regulator of cold stress in *Arabidopsis thaliana* by regulating cold response genes, maintaining reactive oxygen species (ROS) activity, and glucose metabolism (*Liu et al., 2022*). In *Arabidopsis*, the lncRNA *COOLAIR* inhibits the expression of the *FLC* gene under cold stress *via* the reduction of H3K36 methylation and trimethylation of histone H3 lysine 27 (H3K27me3). This inhibition, in *Vitis vinifera* L, confers cold tolerance by regulating hormone signal transduction, secondary metabolite biosynthesis, sucrose metabolism pathways, and various transcription factors, including CBF, WRKY, and NACs. Additionally, in Cassava, lncRNAs like *CRIR1* interact with the *MeCSP5* gene (cold shock protein 5 encoding gene) to help confer cold tolerance, while in *Ammopiptanthus nanus*, the lncRNA *TCONS00065739*

targets *miR530*, contributing to cold stress adaptation by regulating the *TZP* gene (*Jha et al., 2023*). In cotton, the nucleus-localized lncRNA973 enhances salt tolerance by modulating reactive oxygen species (ROS), whereas its knockdown reduces tolerance, leading to plant wilting and leaf yellowing under salt stress (*Zhang et al., 2019*).

However, the role of transposon-derived lncRNAs (TElncRNAs) in response to cold stress remains largely unknown. Transposable elements (TEs), known as mobile DNA, occupy a significant portion of eukaryotic genomes, contribute to genome evolution, and regulate gene expression (*Gebrie, 2023*; *Ramakrishnan et al., 2022*). TEs provide *cis*-regulatory regions, such as promoters and enhancers, that influence both TE-encoded and host gene expression. Many miRNAs and lncRNAs are derived from TEs and play crucial roles in regulatory functions, including target mRNA binding (*Gebrie, 2023*). Additionally, TEs often exhibit tissue-specific functions, contributing to stress tolerance regulation. For instance, tissue-specific and stress-induced TE-lncRNAs (long intergenic noncoding RNAs) have been identified under salt, abscisic acid (ABA), and cold treatments in *Arabidopsis*, rice, and maize. Notably, these TE-lncRNAs are predominantly derived from retrotransposons rather than DNA transposons (*Wang et al., 2017*). Furthermore, TE-lncRNAs, due to their transposon-derived nature, are recognized as transposons themselves and are consequently subjected to similar epigenetic silencing mechanisms. These lncRNAs, like transposons, have the potential to move and propagate within the genome (*Kornienko et al., 2023*).

In Moso bamboo, several lncRNAs have been identified that are associated with abiotic stress, nitrogen metabolism, and secondary cell wall biosynthesis (*Ding et al., 2024*; *Ding et al., 2022*; *Wang et al., 2021*; *Yuan et al., 2022*). Moso bamboo's genome comprises over 63% TEs and thus heavily relies on TE genetic components for its development and stress tolerance. TEs are known for their crucial roles in genome modification (*Guo et al., 2019*; *Liufu et al., 2023*; *Zhao et al., 2018*). In our previous study, we identified differentially methylated and expressed TE-lncRNAs under cold, heat, UV, and salt stress conditions (*Ding et al., 2024*). Despite this, the role of lncRNAs derived from long terminal repeat (LTR) retrotransposons (TElncRNAs) in response to cold stress remains largely unexplored.

To address this gap, our study investigated the function of a novel LTR retrotransposon-derived lncRNA, *Pe-TElncRNA2*, under cold stress in Moso bamboo. *Pe-TElncRNA2* is 616 bp in length and consists of two exons. It was overexpressed in Moso bamboo protoplasts and *Arabidopsis thaliana*. For the functional analysis of *Pe-TElncRNA2*, various experimental approaches were employed, including quantitative real-time PCR (qRT-PCR) for gene expression analysis and assays to evaluate antioxidant enzyme activity. Our results show that *Pe-TElncRNA2* modulates gene-specific expression, enhances antioxidant activities, and improves cold tolerance in the transgenic *Arabidopsis*. These findings suggest that *Pe-TElncRNA2* is a promising candidate for developing cold-resistant crops.

## MATERIALS & METHODS

### Plant materials and growth conditions

Moso bamboo seeds from a single maternal plant and *Arabidopsis thaliana* seeds (Columbia ecotype) were used in this study. The Moso bamboo seeds were collected from the same

plant located in Lingchuan County, Guilin City, Guangxi Province, China. All seedlings were cultivated in a controlled greenhouse under a 16-hour light/8-hour dark photoperiod at 25 °C/22 °C (day/night) and 60% relative humidity.

### Cold stress treatment

For cold stress treatment, five-leaf-stage Moso bamboo seedlings were subjected to 4 °C for 8, 16, 24, and 32 h in a plant incubator. For *Arabidopsis*, four-week-old wild-type and transgenic *Arabidopsis* plants were subjected to 4 °C for 2 and 4 days in a plant incubator. After the respective cold stress treatments, five mature leaves from Moso bamboo seedlings and *Arabidopsis* leaves were collected, immediately frozen in liquid nitrogen, and stored at −80 °C for subsequent RNA extraction. Three biological replicates of each stress treatment were used for both species.

### Identification of *Pe-TElncRNA2*

*Pe-TElncRNA2* and its target genes (*PeFZR2*, *PeNOT3*, *PeABCG44*, and *PeAGD6*) were identified from Moso bamboo transcriptome data generated under cold stress conditions (*Ding et al., 2022*). Notably, all these transcripts exhibited differential expression under cold stress. To further characterize *Pe-TElncRNA2*, we employed several bioinformatic tools, including the Coding-Non-Coding Index (CNCI) (*Sun et al., 2013*), the Coding Potential Calculator 2 (CPC2) (*Kong et al., 2007*), the Pfam protein families database (*Finn et al., 2016*), and PLEK, a predictor of long non-coding RNAs and messenger RNAs based on an improved k-mer scheme (*Li, Zhang & Zhou, 2014*) to analyze the translation potential of *Pe-TElncRNA2*. Additionally, NCBI BLASTN was used to determine Pe-TElncRNA2 sequence similarity to known plant transcripts.

### RNA extraction, cDNA synthesis, and qRT-PCR analysis

Total RNA was extracted from Moso bamboo and *Arabidopsis* samples using the SteadyPure Universal RNA Extraction Kit II (Accurate Biology, Changsha, China), following the manufacturer's protocol. The cDNA synthesis was performed using the Hifair® II 1st Strand cDNA Synthesis SuperMix for qPCR (gDNA digester plus) (Yeasen Biotechnology, Shanghai, China) according to the manufacturer's instructions. Quantitative real-time PCR (qRT-PCR) amplification was conducted on a CFX96 Touch Real-Time PCR System (Bio-Rad) using the Hieff® qPCR SYBR® Green Master Mix (Yeasen Biotechnology, Shanghai, China). Gene-specific primers for *Pe-TElncRNA2*, its target genes, and their *Arabidopsis* homologues were designed using Primer 5.0 software. *AtACTIN2* and *PeNTB* served as reference genes for *Arabidopsis* and Moso bamboo, respectively, while *PeACT* and *PeEFlα* were used as reference genes for cellular localization studies in Moso bamboo. Relative gene expression levels were calculated using the $2^{-\Delta\Delta Ct}$ method. The primer names and their sequences are listed in Table S1.

### Nuclear and cytoplasmic protein extraction from Moso bamboo leaves

For the cellular localization analysis of *Pe-TElncRNA2*, Moso bamboo leaves were cut into the smallest possible pieces. Nuclear and cytoplasmic proteins were extracted using a Nuclear and Cytoplasmic Protein Extraction Kit (Beyotime Biotechnology, Shanghai,
China) and phenylmethylsulfonyl fluoride (PMSF). The cut leaf tissue was homogenized at a ratio of 3:20, and the extraction process was carried out according to the manufacturer's instructions. The extracted proteins were stored at $-80\,°\text{C}$ for future use. RNA extraction, cDNA synthesis, and qRT-PCR analysis were performed as previously described. *PeACT* and *PeEF1A* were used as reference markers for nuclear and cytoplasmic localization, respectively. The primer names and their sequences are listed in Table S1. The formulae used to calculate the relative expression levels in the nucleus and cytoplasm are as follows: Nuclear $\% = 2^{-\Delta CT}$ nucleus/$(2^{-CT}$ cytoplasm $+ 2^{-CT}$ nucleus$)$, Proton $\% = 1 -$ Nuclear $\%$.

## Plasmid construction for *Pe-TElncRNA2* overexpression

To overexpress *Pe-TElncRNA2* in Moso bamboo protoplasts and *Arabidopsis*, the full-length cDNA of *Pe-TElncRNA2* (containing only exons) was amplified by PCR using $2\times$ Hieff Canace® Plus PCR Master Mix (with Dye) (Yesen, Shanghai, China) and a pair of specific primers: *Pe-TElncRNA2*-full length-F and *Pe-TElncRNA2*-full length-R. The amplified *Pe-TElncRNA2* fragments were then cloned into two different vectors: *pUBQ10*, which is driven by the Ubiquitin (UBQ) promoter, and *pER8*, which is driven by the cauliflower mosaic virus (CaMV) 35S promoter. The resulting constructs, *pUBQ10-Pe-TElncRNA2* and *pER8-Pe-TElncRNA2*, were then used for genetic transformation of Moso bamboo protoplasts and *Arabidopsis*, respectively. The primer names and their sequences are listed in Table S1.

## Moso bamboo protoplast genetic transformation

Moso bamboo protoplast isolation and the polyethylene glycol (PEG)-mediated genetic transformation methods were conducted as described in our previous study (*Yu, Ding & Zhou, 2023*). Briefly, 21-day-old Moso bamboo leaf sheaths were cut into the smallest possible pieces and incubated in an enzyme solution for 4 h. Following centrifugation, $2 - 3 \times 10^4$ protoplasts were resuspended in 200 µL of MMG solution (4 mM MES-KOH (pH 5.7), 0.4 M mannitol, and 15 mM $MgCl_2$). Next, a mixture of 10 µg *pUBQ10-Pe-TElncRNA2* and 100 µL protoplasts was incubated with PEG solution (40% PEG4000, 0.8 M mannitol, and 1 M $CaCl_2$) for 4 min. Subsequently, W5 solution (4 mM MES-KOH (pH 5.7), 0.5 M mannitol, and 20 mM KCl) was added to the sample, and the protoplasts were then incubated and harvested. Transfection efficiency of the *pUBQ10-Pe-TElncRNA2* construct in Moso bamboo protoplasts was assessed by confocal laser scanning microscopy (CLSM, Zeiss LSM510; Zeiss, Oberkochen, Germany) after 12–16 h post-transfection.

The experiment was divided into two groups: the protoplasts with the *pUBQ10* empty vector, and the protoplasts with the *pUBQ10-Pe-TElncRNA2* construct. To induce cold stress, the protoplasts from all three groups were incubated at $4\,°\text{C}$ for 4 h in a plant incubator. Subsequently, the protoplasts were centrifuged at 150 g for 2 min, and the supernatant was discarded. RNA extraction, cDNA synthesis, and qRT-PCR analysis were performed immediately as previously described. Three biological replicates were used for gene expression analyses in each group.

### *Arabidopsis* floral dip transformation

*Arabidopsis* transgenic plants were generated using the *Agrobacterium*-mediated floral dip genetic transformation method (*Zhang et al., 2006*). In this process, the *pER8-Pe-TElncRNA2* construct was introduced into *Arabidopsis* plants by dipping fully blossomed inflorescences for 30 s in an *Agrobacterium* suspension containing the construct. The suspension included a 5% sucrose solution and 0.03% (vol/vol) Silwet L-77. The transgenic T1 seeds were harvested from plants grown on half-strength Murashige and Skoog (MS) medium supplemented with 35 mg/L hygromycin. PCR amplification using *Pe-TElncRNA2*-specific primers was employed to screen for the transgenic plants.

### Measurement of proline content

To assess antioxidant activities in *Arabidopsis* transgenic plants under cold stress conditions, proline content was measured using kits from Suzhou Comin Biotechnology Co., Ltd., Suzhou, China, following the manufacturer's instructions. Briefly, after the cold stress treatment, fresh *Arabidopsis* transgenic leaves (0.05 g) from each collected sample were homogenized in an ice bath using a TGrinder tissue grinder (China). The proline extraction was then performed, and the absorbance data were recorded. Finally, the proline content was calculated according to the supplier's instructions.

### Superoxide dismutase activity analysis

Superoxide dismutase (SOD) activity was assessed using an activity assay kit from Suzhou Comin Biotechnology Co., Ltd., Suzhou, China, following the manufacturer's instructions. Briefly, after the cold stress treatment, fresh *Arabidopsis* transgenic leaves (0.05 g) from each collected sample were homogenized in one mL of extraction buffer. The homogenates were then centrifuged at 12,000 rpm at 4 °C for 10 min to obtain a crude enzyme extract, and the supernatant was used for SOD enzyme activity analysis. The absorbance data at $\lambda = 560$ nm were recorded. SOD enzyme activity was defined as a unit of enzyme activity (U/mL) when the percentage of inhibition in the reaction system reached 50%.

### Determination of malondialdehyde content

To evaluate the oxidative stress level in *Arabidopsis* transgenic plants under cold stress conditions, malondialdehyde (MDA) content—a lipid peroxidation marker and indicator of oxidative stress—was measured using a kit from Suzhou Comin Biotechnology Co., Ltd., Suzhou, China, according to the manufacturer's instructions. Briefly, after the cold stress treatment, fresh *Arabidopsis* transgenic leaves (0.05 g) from each collected sample were homogenized in an ice bath using a TGrinder tissue grinder (Tiangen, Beijing, China). The MDA extraction was then carried out following the supplier's instructions. The corresponding absorbance data were recorded, and the MDA content was calculated.

### Determination of photosynthetic efficiency

To assess photosynthetic efficiency in *Arabidopsis* transgenic plants under cold stress conditions, the maximum photochemical efficiency PS II ($F_v/F_m$) was measured in dark-adapted leaves using the Multi-Function Plant Efficiency Analyser (M-PEA). After dark-adapting the leaves for 20 min, the fluorescence parameters were measured. In this process,

$F_0$ represents the initial fluorescence, $F_m$ is the maximum fluorescence, and $F_v/F_m$ is calculated as $(F_m - F_0)/F_m$.

## Statistical analysis

Pearson correlation coefficients were calculated to assess the relationship between the relative expression levels of *Pe-TElncRNA2* and its target genes. The data are presented as mean values of three replicates ± standard deviation. Fisher's least significant difference (LSD) and Duncan's multiple range test (DMRT) were used for multiple comparisons among the samples, with analysis performed using GraphPad Prism 8.0 (GraphPad Software, San Diego, CA, USA, http://www.graphpad.com). Treatment means followed by different lowercase letters are significantly different at $p < 0.05$.

## RESULTS

### Structural analysis of *Pe-TElncRNA2*

Our previous studies revealed that approximately 12.4% of lncRNAs in Moso bamboo originated from TEs, primarily from the Ty1/*copia* and Ty3/*gypsy* LTR retrotransposon superfamilies. While TE-lncRNAs and non-TE-lncRNAs exhibited similar length distributions and expression patterns, TE-lncRNAs displayed stress-specific expression profiles, often downregulated under cold stress, and were predicted to target multiple genes. Moreover, epigenetic regulation, particularly promoter and genic region methylation, was found to suppress lncRNA expression (*Ding et al., 2024*; *Ding et al., 2022*). Building upon these findings, the current study delved deeper into the functional characterization of a novel LTR retrotransposon-derived lncRNA, *Pe-TElncRNA2*, and its target genes within the context of cold stress in Moso bamboo, utilizing the same transcriptome dataset (*Ding et al., 2022*).

Pe-TElncRNA2 is a 616 bp full-length lncRNA composed of two exons and lacking protein-coding potential, as confirmed by CNCI, Pfam, and PLEK analyses. While CPC2 predicted a low coding probability (0.005) and Fickett score (0.411), the overall evidence strongly supports its classification as a lncRNA. It is located within an LTR retrotransposon on Chromosome PH01004968, spanning positions 16,954–17,569 (Fig. 1A). BLASTN analysis has further confirmed that *Pe-TElncRNA2* is a novel lncRNA with no sequence similarity to any known plant transcripts.

### The expression profiles of *Pe-TElncRNA2* and its related genes

Based on the transcriptome dataset, *Pe-TElncRNA2* was upregulated under cold stress. To further investigate, *Pe-TElncRNA2* was analyzed in Moso bamboo seedlings using qRT-PCR. *Pe-TElncRNA2* expression peaked at 8 h of cold treatment (4 °C) and was significantly higher in the stems compared to the leaves, roots, and buds (Fig. 1B). Further cellular localization analysis showed that *Pe-TElncRNA2* expression was significantly higher in the cytoplasm, with 72.53% expression in the cytoplasm and 27.47% in the nucleus. Conversely, the reference gene *PeEF1α* was expressed entirely in the cytoplasm, while *PeACT* was significantly higher in the nucleus than in the cytoplasm (Fig. 1C). These results indicate that *Pe-TElncRNA2* mainly plays a role in the cytoplasm.

Peer J

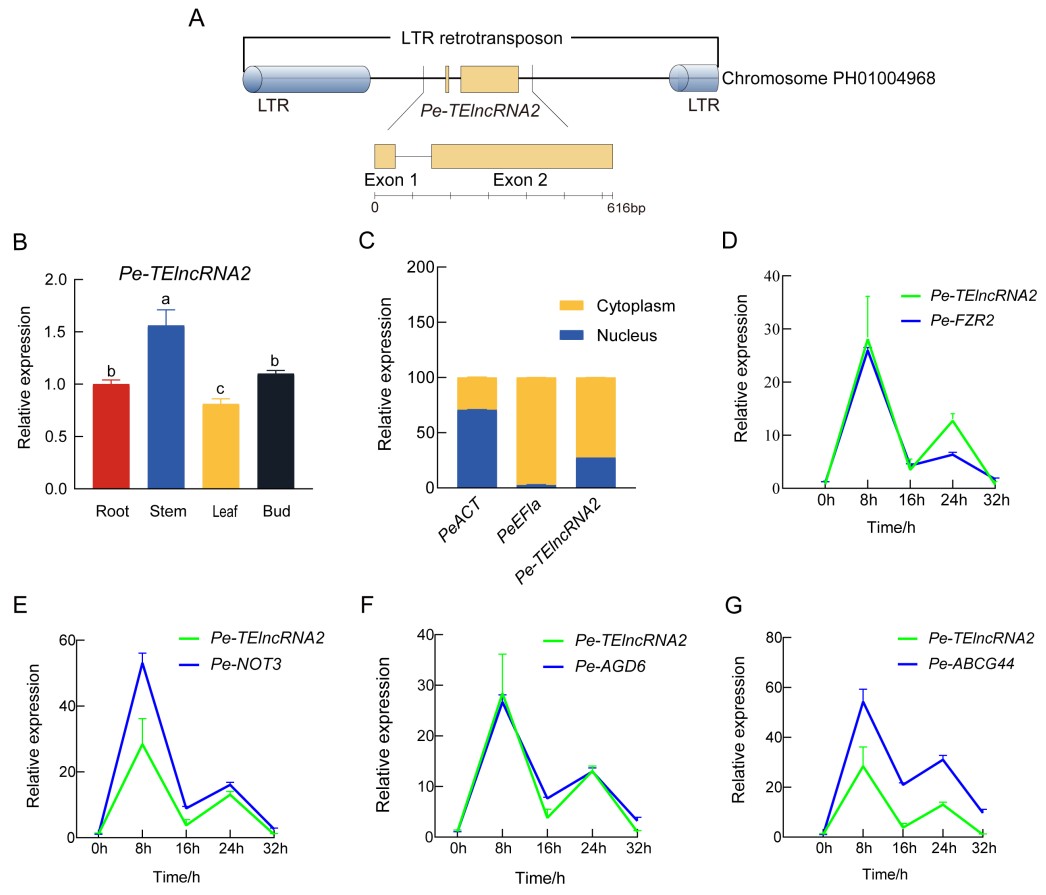

**Figure 1  Structural analysis of *Pe-TElncRNA2*, its expression profile, and its target genes in Moso bamboo.** (A) The genomic location of the 616 bp full-length *Pe-TElncRNA2* within an LTR retrotransposon on chromosome PH01004968. The exons, lacking coding potential, are indicated by green rectangles. (B) Tissue-specific expression of *Pe-TElncRNA2* in the root, stem, leaf, and bud. (C) Cellular localization of *Pe-TElncRNA2* and the reference genes (*PeACT* and *PeEF1α*) in the cytoplasm and nucleus. (D-G) Expression patterns of *Pe-TElncRNA2* and four genes, such as *PeFZR2*, *PeNOT3*, *PeABCG44*, and *PeAGD6*, in the leaves under cold stress. Under cold treatment, *Pe-TElncRNA2* and four genes exhibited a biphasic expression pattern, peaking first at 8 h, then declining, reaching a second peak at 24 h, and decreasing again.

Further analysis revealed that *Pe-TElncRNA2* acts as a cold-induced regulator of four genes: fizzy-related 2 protein (*FZR2*), negative regulator of transcription subunit 3 (*NOT3*), ABC transporter G family member 44 (*ABCG44*), and ADP-ribosylation factor GTPase-activating protein (*AGD6*). To investigate their expression dynamics, *Pe-TElncRNA2* and its target genes (*PeFZR2*, *PeNOT3*, *PeABCG44*, and *PeAGD6*) were analyzed using qRT-PCR in Moso bamboo seedlings subjected to cold treatment for 8, 16, 24, and 32 h. The expression patterns of the target genes exhibited a strong correlation with that of *Pe-TElncRNA2* (Figs. 1D–1G). Pearson correlation analysis further revealed correlation coefficients greater than 0.8 between *Pe-TElncRNA2* and its target genes (Fig. S1). The expression patterns of *Pe-TElncRNA2* and its regulated genes exhibited a biphasic response

to cold stress, with significant upregulation at 8 and 24 h. The expression levels decreased between these peak times and further declined with extended cold exposure (Figs. 1D–1G). These findings suggest that *Pe-TElncRNA2* is involved in the cold stress response of Moso bamboo by regulating the expression of these genes.

### *Pe-TElncRNA2* overexpression in Moso bamboo protoplasts

To investigate the regulatory function of *Pe-TElncRNA2* on these four genes under cold stress, a *pUBQ10-Pe-TElncRNA2* overexpression vector (Fig. S2A) was constructed and transfected into Moso bamboo protoplasts using the polyethylene glycol (PEG)-mediated method. This achieved a transgenesis success rate of up to 50% transformation efficiency. The transfected protoplasts were subsequently subjected to a 4-hour cold treatment. *Pe-TElncRNA2* expression was significantly elevated in the *pUBQ10-Pe-TElncRNA2*-transfected protoplasts compared to the control, *pUBQ10* empty vector-transfected protoplasts, under cold stress conditions (Fig. S2B). Concurrently, the expression levels of the target genes (*PeFZR2*, *PeNOT3*, *PeABCG44*, and *PeAGD6*) were significantly elevated in the *pUBQ10-Pe-TElncRNA2*-transfected protoplasts compared to the control under cold stress conditions (Fig. 2). These findings suggest that *Pe-TElncRNA2* positively regulates these four genes in response to cold stress in Moso bamboo.

### *Pe-TElncRNA2* overexpression in *Arabidopsis*

To investigate the phenotypic effects of *Pe-TElncRNA2*, its full-length cDNA (only exons) was cloned into the overexpression vector *pER8*. The resulting construct (*pER8-Pe-TElncRNA2*) was introduced into *Arabidopsis via Agrobacterium*-mediated floral dip transformation (*Zhang et al., 2006*). Three independent transgenic *Arabidopsis* lines were generated through PCR screening (Fig. S3). Under cold stress, the transgenic plants exhibited less water loss and shrinkage and were greener compared to the wild-type plants after 2 days of treatment. Nonetheless, both transgenic and wild-type plants exhibited similar phenotypic responses, including leaf base purpling and leaf crumpling due to water loss, when subjected to 4-day cold treatments (Fig. 3). However, *Pe-TElncRNA2* expression levels were significantly higher in the transgenic plants compared to the wild-type plants after 2 days of cold treatment, but there was no significant difference after 4 days of cold treatment (Fig. 4). These results suggest that *Pe-TElncRNA2* plays an important role in the early cold stress response of *Arabidopsis*.

Concurrently, the expression levels of the genes (AT4G22910 (*AtFZR2*), AT5G18230 (*AtNOT3*), AT5G18230 (*AtABCG44*), and AT1G53710 (*AtAGD6*)), which are homologous to *PeFZR2*, *PeNOT3*, *PeABCG44*, and *PeAGD6*, respectively were also significantly higher in the transgenic plants compared to the wild-type plants after 2 days of cold treatment. The result suggest that *Pe-TElncRNA2* also regulates Moso bamboo's homologous genes in *Arabidopsis*, contributing to early cold stress responses.

### Antioxidant activities and photosynthetic efficiency

To assess antioxidant capacity and photosynthetic efficiency, SOD activity, proline content, MDA content, and maximum photochemical efficiency ($F_v/F_m$) in dark-adapted leaves were measured in both transgenic and wild-type *Arabidopsis* plants following cold treatment.

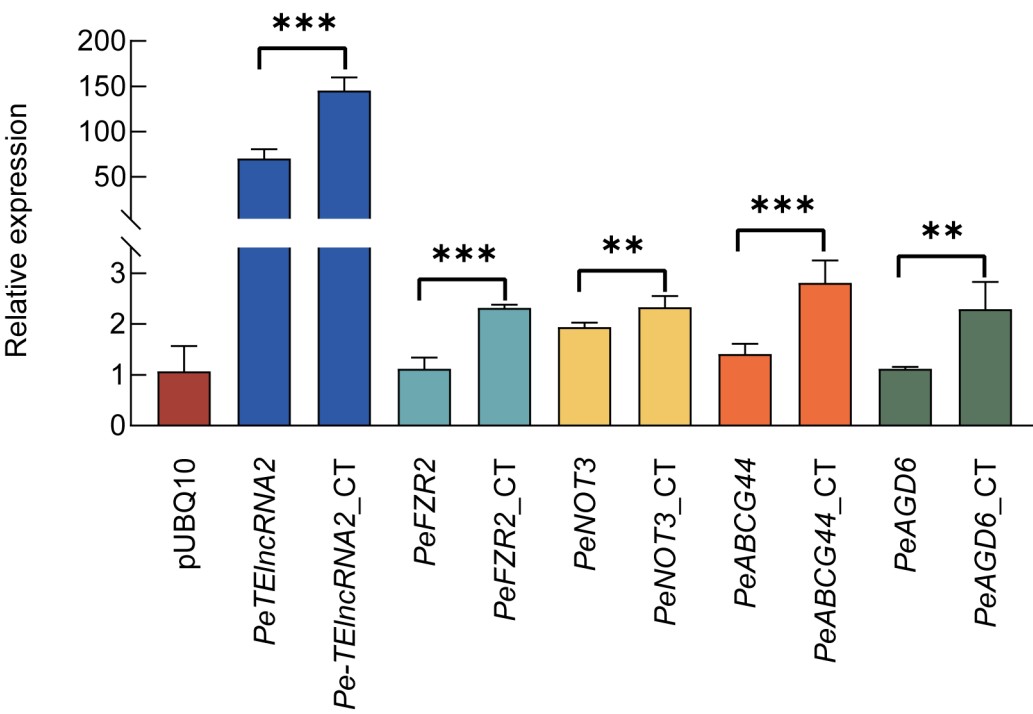

**Figure 2 Relative expression levels of *Pe-TElncRNA2* and four genes in the *pUBQ10-Pe-TElncRNA2*-transfected Moso bamboo protoplasts.** The expression levels of *Pe-TElncRNA2* (A) *PeFZR2* (B) *PeNOT3* (C) *PeABCG44* (D) and *PeAGD6* (E) in the pUBQ 10-Pe-TElncRNA2 -transfected protoplasts under cold stress conditions after 4 h of treatment. *$p < 0.05$, **$p < 0.01$, and ***$p < 0.001$ indicate statistically significant differences. CT indicates cold stress conditions after 4 h of treatment. The empty vector *pUBQ10* and *pUBQ10-Pe-TElncRNA2* -transfected protoplasts under normal conditions served as controls.

Compared to the wild-type controls, the transgenic plants exhibited enhanced antioxidant capacity and photosynthetic efficiency under cold stress (Fig. 5). SOD activity and proline content were significantly increased in the transgenic plants, while MDA content, a lipid peroxidation marker (a marker of oxidative stress), was reduced compared to the wild-type. Additionally, the $F_v/F_m$ ratio significantly decreased in both wild-type and transgenic plants under cold stress; however, the transgenic plants maintained higher $F_v/F_m$ values, indicating improved photoprotection under cold stress conditions. These results suggest that *Pe-TElncRNA2* overexpression enhances cold stress tolerance in *Arabidopsis*.

## DISCUSSION

### Identification of a novel *TElncRNA2* in Moso bamboo

Moso bamboo is a high-yielding, fast-growing crop with significant economic value in food and construction. However, its cultivation is largely limited to tropical and subtropical regions due to its sensitivity to cold temperatures (*Chen et al., 2022*). Additionally, a significant negative correlation between monthly precipitation and internode diameter and thickness has been observed, particularly during the colder months of December

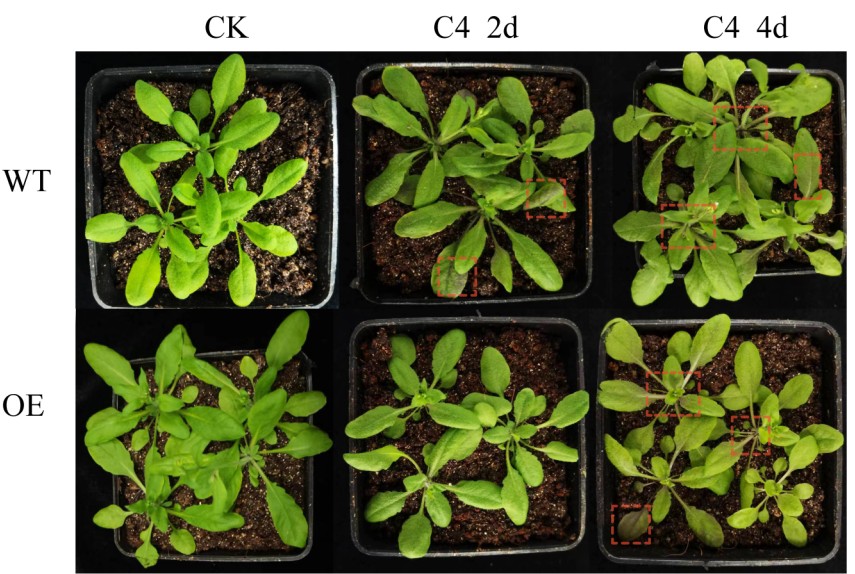

**Figure 3 Phenotypic comparison between the wild-type and transgenic *Arabidopsis* under cold stress.** Representative images show the wild-type (WT) and transgenic *Arabidopsis* plants overexpressing *Pe-TElncRNA2* , subjected to 2-day (C4-2d) and 4-day (C4-4d) cold treatments. Transgenic plants exhibited increased plant vigor, as indicated by reduced water loss and yellowing, compared to WT under cold stress conditions.

and January (*Zhang et al., 2024*). This correlation affects primary thickening growth and, consequently, internode size, which impacts its global production and utilization.

Cold stress inhibits cell division-related genes (*PeCYC1BAT*, *PeCYCB2;4*, *PeCDC20.1*, *PeMAD2*, *PeTPX2*, and *PeTCX2*) and cell elongation-related genes (*PeEXPA1*) during the rapid growth period in Moso bamboo (*Chen et al., 2022*). Given the importance of cold stress tolerance, several studies have identified cold-related genes in Moso bamboo, including *PeEREBP*, *PeHSF*, *PeMYB*, *PeNAC, PeWRKY*, and *PeLEA* (*Huang et al., 2022*; *Liu et al., 2020*; *Wang et al., 2022*). However, most of these genes have yet to be validated. Additionally, non-coding RNAs, including lncRNAs, play a role in regulating stress responses in Moso bamboo (*Ding et al., 2022*; *Yu, Ding & Zhou, 2023*), underscoring the complexity of its stress response mechanisms. Despite this, the role of lncRNAs in response to cold stress remains elusive, particularly the function of transposon-derived lncRNAs (TElncRNAs), which is still largely unknown.

Therefore, this study provides the first evidence of the role of TElncRNAs in cold stress response in Moso bamboo. We identified a novel TElncRNA, *Pe-TElncRNA2*, located within an LTR retrotransposon on chromosome PH01004968 (Fig. 1A). This lncRNA, which is 616 bp in length and composed of two exons, has no translation potential and lacks evolutionary conservation, as confirmed by CNCI, Pfam, and PLEK analyses. Previous studies have demonstrated that lncRNAs constitute 22.9%, 49.7%, and 51.5% of the total transcriptome in *Arabidopsis*, rice, and maize, respectively. Retrotransposons play a crucial role in shaping plant stress responses by influencing the epigenome. For

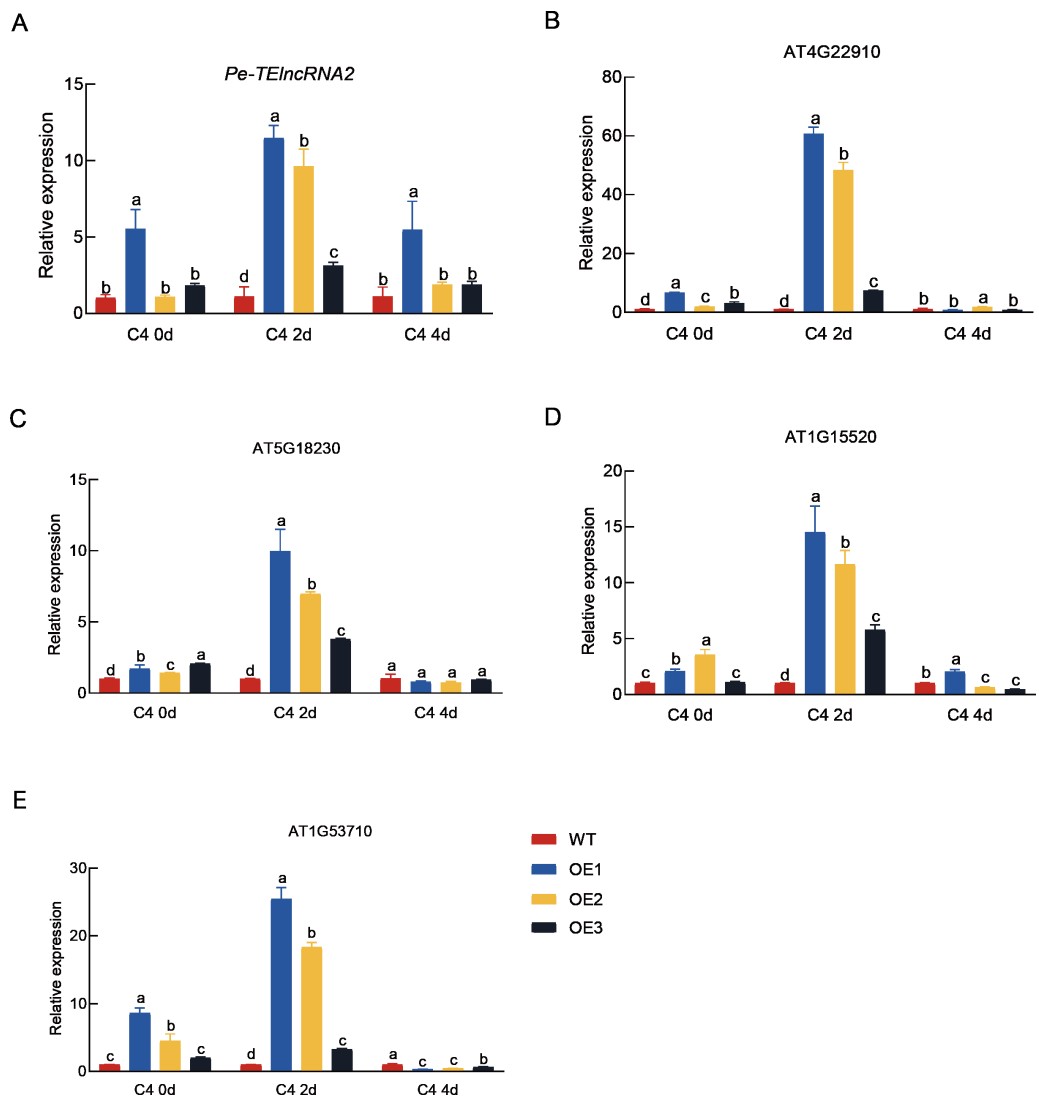

**Figure 4  Expression analysis of *Pe-TElncRNA2*. and four homologous genes in the wild-type and transgenic *Arabidopsis* under cold stress.** Relative expression levels of *Pe-TElncRNA2* (A), *AT4G22910* (B), *AT5G18230* (C), *AT1G1552* 0 (D), and *AT1G53710* (E) in the transgenic *Arabidopsis* lines (OE1, OE2, and OE3) overexpressing *Pe-TElncRNA2* , exposed to 2-day (C4-2d) and 4-day (C4-4d) cold treatments. The four genes, including *AT4G22910* (*AtFZR2*), *AT5G18230* (*AtNOT3*), *AT5G18230* (*AtABCG44*), and *AT1G53710* (*AtAGD6*), are homologous to *PeFZR2*, *PeNOT3*, *PeABCG44*, and *PeAGD6*, respectively. *AtACTIN2* was used as the reference gene, and the samples with 0-hour cold treatment served as control. The data represent the mean values of three replicates ± standard deviation. Fisher's least significant differences (LSD) and Duncan's multiple range test (DMRT) were used for multiple comparisons among all the samples. Different lowercase letters indicate significant differences at $p < 0.05$.

example, *Arabidopsis*, rice, and maize predominantly generate stress-induced TE-lncRNAs from retrotransposons (*Wang et al., 2017*).

In Moso bamboo, TElncRNAs also predominantly originate from LTR retrotransposons and exhibit stress-specific expression patterns. Typically, these lncRNAs are downregulated under cold and salt stress conditions, while upregulation is observed under heat stress.

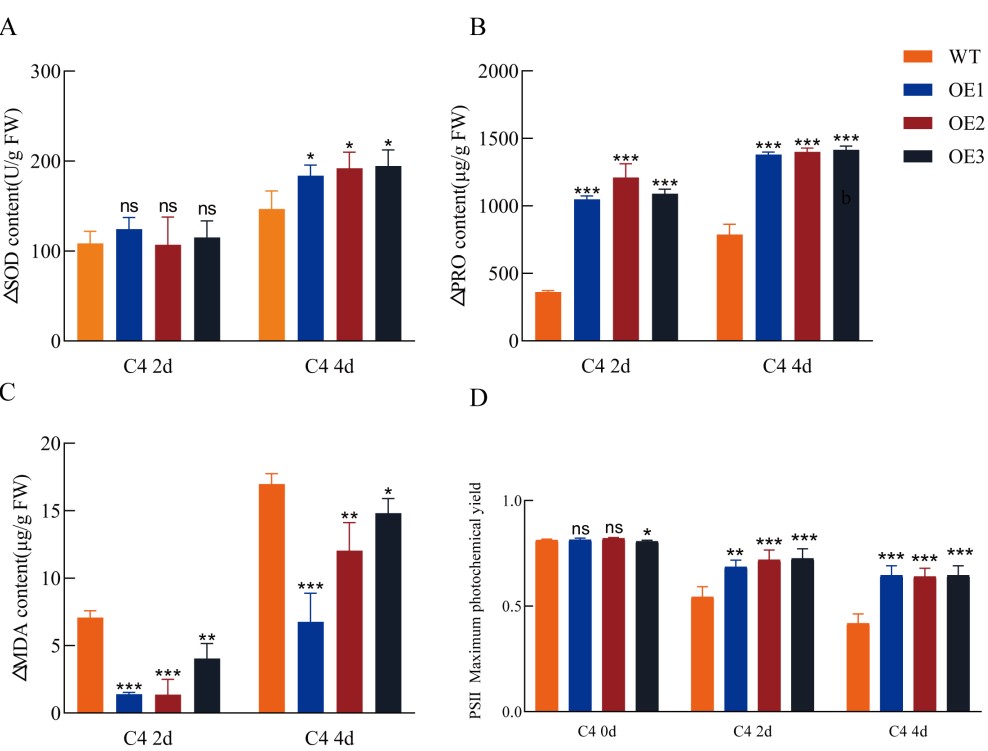

**Figure 5  Antioxidant enzyme activity, osmolyte accumulation, lipid peroxidation, and photosynthetic efficiency in transgenic and wild-type (WT) *Arabidopsis* under cold stress.** Superoxide dismutase (SOD) activity (A), proline content (B), malondialdehyde (MDA) content (C), and chlorophyll content (D). Increased levels of SOD and proline, along with an increased maximum quantum efficiency of photosystem II (PSII) ($F_v/F_m$ ratio) and decreased MDA content in the leaves, indicate an enhanced cold stress response. OE1, OE2, and OE3 are the transgenic lines overexpressing *Pe-TElncRNA2*. FW represents the fresh weight of leaf tissues. The data represent the mean values of three replicates $\pm$ standard deviation. Fisher's least significant differences (LSD) and Duncan's multiple range test (DMRT) were used for multiple comparisons among all the samples. $^*p < 0.05$, $^{**}p < 0.01$, and $^{***}p < 0.001$ indicate statistically significant differences.

For instance, *Pe-TElncRNA3* plays a crucial role in the heat stress response by regulating downstream genes (*PH02Gene25732*, *PH02Gene25729*, and *PH02Gene03426*) (*Ding et al., 2024*). In contrast, our study found that *Pe-TElncRNA2*, also derived from an LTR retrotransposon, is upregulated under cold stress in Moso bamboo (Figs. 1B–1G). This highlights a broader trend in which retrotransposons contribute to stress adaptation through the regulation of lncRNAs.

Additionally, lncRNA expression varies across different tissues and developmental stages. For instance, *AtR8* and *BoNR8* lncRNAs are predominantly expressed in the root and root elongation zone epidermis of *Arabidopsis*, respectively (*Wu et al., 2019*; *Wu et al., 2012*). In contrast, *Pe-TElncRNA2* is specifically upregulated in the stem under cold stress conditions, suggesting that lncRNA tissue-specific expression patterns can vary across different organs in Moso bamboo.

### Transient overexpression of *Pe-TElncRNA2* in Moso bamboo

The subcellular localization of lncRNAs can significantly influence their function. For example, nuclear lncRNAs often participate in pre-rRNA transcriptional regulation (*Xing et al., 2017*), while cytoplasmic lncRNAs modulate mRNA stability, translation, signaling pathways, and interact with RNA-binding proteins to regulate gene expression (*Statello et al., 2021*). In this study, *Pe-TElncRNA2* was predominantly localized in the cytoplasm (Fig. 1C and Fig. S2B), suggesting its potential role in mRNA stability, translation, and post-transcriptional regulation.

Due to the absence of a stable transformation system for Moso bamboo, we employed protoplast transformation to overexpress *Pe-TElncRNA2*. Given the current limitations, protoplasts offer a viable alternative for assessing gene functionality Moso bamboo (*Yu, Ding & Zhou, 2023*). This approach allowed us to verify its association with downstream genes through transient expression without altering Moso bamboo's genome. Notably, the upregulation of *Pe-TElncRNA2* was associated with the concurrent upregulation of its regulating genes (*PeFZR2*, *PeNOT3*, *PeABCG44*, and *PeAGD6*) (Fig. 2). This observation is in line with our previous study, where upregulation of *Pe-lncRNA1* coincided with the upregulation of its regulating genes (such as *PH02Gene33364*, *PH02Gene38550*, *PH02Gene43330*, *PH02Gene19065*, *PH02Gene05460*, *PH02Gene26812*, *PH02Gene35897*, and *PH02Gene50461*) under UV-B stress (*Yu, Ding & Zhou, 2023*). Our previous study demonstrated that, without genome integration, LTR retrotransposons increase their copies under heat stress (*Papolu et al., 2021*). This finding aligns with the current study, where LTR retrotransposon-derived *Pe-TElncRNA2* is expressed under cold stress conditions. This supports the hypothesis that TE-lncRNAs, as transposons, have the potential to move and propagate within the genome (*Kornienko et al., 2023*).

### Stable overexpression of *Pe-TElncRNA2* in *Arabidopsis*

Given the potential regulatory role of *Pe-TElncRNA2* in cold stress response in Moso bamboo, we validated its stable function by overexpressing it in *Arabidopsis*. Consistent with our protoplast transient expression results, the stable overexpression of *Pe-TElncRNA2* in *Arabidopsis* demonstrated similar results (Fig. 4). The transgenic plants exhibited reduced water loss, maintained leaf color, and showed increased expression of *Pe-TElncRNA2* and its regulating genes, *AtFZR2*, *AtNOT3*, *AtABCG44*, and *AtAGD6*. These findings suggest that *Pe-TElncRNA2* positively regulates homologous genes in both Moso bamboo and *Arabidopsis*, thereby contributing to early cold stress responses. This observation aligns with our previous study, where *Pe-lncRNA1* upregulation also positively regulated homologous genes in both species under UV-B stress (*Yu, Ding & Zhou, 2023*).

### *Pe-TElncRNA2*f-modulated antioxidant activities and photosynthetic efficiency

To elucidate the mechanisms underlying *Pe-TElncRNA2*-mediated cold tolerance, we assessed antioxidant capacity by measuring SOD activity, proline content, and MDA levels, as well as photosynthetic efficiency through $F_v/F_m$ measurements (Fig. 5). The transgenic plants exhibited significantly increased SOD activity and proline content, while

MDA levels were reduced compared to the wild-type controls. Moreover, the transgenic plants maintained higher $F_v/F_m$ values, indicating improved photoprotection under cold stress conditions. These findings align with previous studies demonstrating that enhanced antioxidant capacity and photosynthetic efficiency contribute to cold stress tolerance (*Ghosh et al., 2024*; *Xu et al., 2023*).

*Pe-TElncRNA2*-induced regulation of antioxidant enzyme genes contributes to this enhanced antioxidant capacity. Previous studies have demonstrated that lncRNAs can modulate antioxidant responses. For instance, lncRNA *CIL1* maintains ROS homeostasis through cold response genes (*Liu et al., 2022*), while *lncRNA973* enhances salt tolerance by regulating ROS scavenging (*Zhang et al., 2019*). Moreover, lncRNA and mRNA co-expression plays a critical role in signal transduction and stress tolerance (*Cui et al., 2019*; *Tan et al., 2020*). For example, the lncRNA *MtCIR2* and its target genes *MtCBF/DREB1s* regulate freezing tolerance in *Medicago truncatula*. *MtCIR2* overexpression upregulates the expression of these target genes, while mutant *MtCIR2* downregulates them (*Zhao et al., 2023*).

In this study, *Pe-TElncRNA2* overexpression upregulates the expression of its downstream genes. The coordinated expression of *Pe-TElncRNA2* and its downstream genes contributes to enhanced cold tolerance by modulating antioxidant activity and photosynthetic efficiency. Given that *Pe-TElncRNA2* originates from an LTR retrotransposon, it is reasonable to conclude that this may act as a *cis*-regulatory element, such as a promoter or enhancer, impacting the expression of antioxidant enzyme genes (*Gebrie, 2023*). These findings suggest that *Pe-TElncRNA2* acts as a key regulator in a complex regulatory network that promotes cold stress resilience in Moso bamboo.

## CONCLUSION

This study uncovered the critical role of a novel LTR retrotransposon-derived lncRNA, *Pe-TElncRNA2*, in enhancing cold tolerance in Moso bamboo. The results demonstrated that *Pe-TElncRNA2* was predominantly localized in the cytoplasm, highly expressed in the stem, and positively regulated the expression of key genes involved in antioxidant defense and photosynthesis. Overexpression of *Pe-TElncRNA2* in *Arabidopsis* conferred increased cold tolerance, characterized by reduced oxidative stress and improved photosynthetic efficiency. These findings collectively suggest that *Pe-TElncRNA2* functions as a crucial regulator in the complex network of cold stress responses in plants. Understanding the molecular mechanisms underlying *Pe-TElncRNA2* function may provide valuable insights for developing cold-tolerant crop varieties.

## ACKNOWLEDGEMENTS

The authors wish to express their gratitude for the help and support provided by the staff at the State Key Laboratory of Subtropical Silviculture and the Institute of Bamboo Research of Zhejiang A&F University. We would like to extend our sincere gratitude and appreciation to all reviewers for their valuable comments.

### Funding

This work was supported by the National Natural Science Foundation of China (Grant No. 32471980) and the Zhejiang Provincial Natural Science Foundation of China (Grant No. LZ24C160002). The funders had no role in study design, data collection and analysis, decision to publish, or preparation of the manuscript.

### Grant Disclosures

The following grant information was disclosed by the authors:
The National Natural Science Foundation of China: 32471980.
The Zhejiang Provincial Natural Science Foundation of China: LZ24C160002.

### Competing Interests

The authors declare there are no competing interests.

### Author Contributions

- Jiamin Zhao conceived and designed the experiments, performed the experiments, analyzed the data, prepared figures and/or tables, authored or reviewed drafts of the article, and approved the final draft.
- Yiqian Ding performed the experiments, analyzed the data, prepared figures and/or tables, authored or reviewed drafts of the article, and approved the final draft.
- Muthusamy Ramakrishnan analyzed the data, authored or reviewed drafts of the article, and approved the final draft.
- Long-Hai Zou conceived and designed the experiments, authored or reviewed drafts of the article, and approved the final draft.
- Yujing Chen performed the experiments, prepared figures and/or tables, and approved the final draft.
- Mingbing Zhou conceived and designed the experiments, authored or reviewed drafts of the article, and approved the final draft.

### Data Availability

The raw data is available in the Supplemental Files.

### Supplemental Information

Supplemental information for this article can be found online at http://dx.doi.org/10.7717/peerj.19056#supplemental-information.

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
