# Peer review of "LTR retrotransposon-derived novel lncRNA2 enhances cold tolerance in Moso bamboo by modulating antioxidant activity and photosynthetic efficiency"

_PeerJ, doi:10.7717/peerj.19056_

## Round 0.1 · original submission · Major Revisions

Dear colleagues, your manuscript has now been assessed by expert reviewers who believe that it has merit for publication following revisions

Reviewer 1 ·

Basic reporting

Comments for manuscript
“LTR retrotransposon-derived novel lncRNA2 enhances cold tolerance in Moso bamboo by modulating antioxidant activity and photosynthetic efficiency.” This study reveals the significant role of the novel LTR retrotransposon-derived lncRNA, Pe-TElncRNA2, in improving cold tolerance in Moso bamboo. The manuscript is clear and logical, making it suitable for publication, though it requires some revisions. The following comments are provided:

Major Points:

Line 198: The manuscript should include a more detailed explanation of Moso bamboo protoplast transformation efficiency, along with specifics on the process and success rates.
Line 314: The description of “yellowing and water loss” after 4 days of cold treatment appears overstated. Please revise the statement to better reflect the observations in Figure 4 or clarify if the figure needs further scrutiny.
Line 159: To enhance data presentation, please provide the formulas used to calculate relative expression levels in the nucleus and cytoplasm.
The introduction lacks sufficient detail on the critical techniques and methodologies employed; please expand on these aspects.
Minor Points:
5. Line 34-35: Correct the grammatical error by changing "regulate" to "regulated" for verb tense consistency.
6. Seed Origin: The geographical origin of the Moso bamboo seeds is unclear; please specify the region from which the seeds were sourced.
7. Line 145: Ensure consistent formatting for "RT-qPCR," introducing the abbreviation correctly on its first use and maintaining consistency thereafter.
8. Lines 197-198: If there is a specific reference or methodology for protoplast treatment, please include it to support the described methodology.
9. Tense Consistency: Review and correct several tense errors throughout the manuscript to ensure grammatical accuracy.

Please address these comments to enhance the clarity and accuracy of the manuscript before submission for publication.

Experimental design

no comment

Validity of the findings

no comment

Additional comments

Comments for manuscript
“LTR retrotransposon-derived novel lncRNA2 enhances cold tolerance in Moso bamboo by modulating antioxidant activity and photosynthetic efficiency.” This study reveals the significant role of the novel LTR retrotransposon-derived lncRNA, Pe-TElncRNA2, in improving cold tolerance in Moso bamboo. The manuscript is clear and logical, making it suitable for publication, though it requires some revisions. The following comments are provided:

Major Points:

Line 198: The manuscript should include a more detailed explanation of Moso bamboo protoplast transformation efficiency, along with specifics on the process and success rates.
Line 314: The description of “yellowing and water loss” after 4 days of cold treatment appears overstated. Please revise the statement to better reflect the observations in Figure 4 or clarify if the figure needs further scrutiny.
Line 159: To enhance data presentation, please provide the formulas used to calculate relative expression levels in the nucleus and cytoplasm.
The introduction lacks sufficient detail on the critical techniques and methodologies employed; please expand on these aspects.
Minor Points:
5. Line 34-35: Correct the grammatical error by changing "regulate" to "regulated" for verb tense consistency.
6. Seed Origin: The geographical origin of the Moso bamboo seeds is unclear; please specify the region from which the seeds were sourced.
7. Line 145: Ensure consistent formatting for "RT-qPCR," introducing the abbreviation correctly on its first use and maintaining consistency thereafter.
8. Lines 197-198: If there is a specific reference or methodology for protoplast treatment, please include it to support the described methodology.
9. Tense Consistency: Review and correct several tense errors throughout the manuscript to ensure grammatical accuracy.

Please address these comments to enhance the clarity and accuracy of the manuscript before submission for publication.

Reviewer 2 ·

Basic reporting

The work by Zhao et.al. has demonstrated the identification of a novel TElncRNA in the cold tolerance of Moso bamboo. The manuscript is well-written with clear introduction to the background and purpose of the study.
However, there are several issues that are essential to the validity of the conclusions, hope the authors could address before considered for publication.

1. The paper stated that Pe-TElncRNA2 acted as a regulator of FZR2, NOT3, ABCG44, and AGD6 genes. However, the data only demonstrated co-expression of those genes rather than a causal relationship. Could the authors provide additional evidence to confirm the regulatory effect of Pe-TElncRNA2 on these genes (for instance, the up-regulation of downstream genes in the lncRNA2 OE line even without cold treatment, or the non-responsiveness to cold treatment of these genes in lncRNA2 knockdown lines)?

2. Figure 2 just showcased that the protoplast transformation method worked, which is probably more suitable as supplementary rather than a main figure.

3. In figure 4, the phenotypic differences were difficult to tell. Could the authors mark the regions more clearly or include quantitative measurements?

4. It was stated in the manuscript that Pe-TElncRNA2 is a novel lncRNA with no sequence similarity to any known. Could the authors offer any explanation why it could regulate the Arabidopsis homologous genes if no similar sequences exist?

Experimental design

NA

Validity of the findings

NA

Additional comments

NA

·

Basic reporting

This study offers a fresh perspective on the relationship between TE and lncRNAs.
The article is clear and well-written.

Experimental design

Statistical analyses were performed and experiments were conducted repeatedly.

Validity of the findings

This work provides a new insight into the connection between TE and lncRNAs.

---

## Round 0.2 · accepted · Accept

Dear colleagues the reviewers have completed their assessment and your manuscript can be accepted

Reviewer 1 ·

Basic reporting

Congratulations, my concerns have been well answered.

Experimental design

Congratulations, my concerns have been well answered.

Validity of the findings

Congratulations, my concerns have been well answered.

Additional comments

Congratulations, my concerns have been well answered.